# Agents of Campylobacteriosis in Different Meat Matrices in Brazil

**DOI:** 10.3390/ijerph19106087

**Published:** 2022-05-17

**Authors:** Micaela Guidotti Takeuchi, Roberta Torres de Melo, Carolyne Ferreira Dumont, Jéssica Laura Miranda Peixoto, Gabriella Rayane Aparecida Ferreira, Mariana Comassio Chueiri, Jocasta Rodrigues Iasbeck, Marcela Franco Timóteo, Bárbara de Araújo Brum, Daise Aparecida Rossi

**Affiliations:** Laboratory of Molecular Epidemiology, Federal University of Uberlândia, Uberlândia 38402-018, Brazil; micaela.guidottitakeuchi@ufu.br (M.G.T.); carolyne.dumont@ufu.br (C.F.D.); jessica.peixoto@ufu.br (J.L.M.P.); gabriella.aparecida@ufu.br (G.R.A.F.); mariana.chueiri@ufu.br (M.C.C.); iasbeck_jocasta@hotmail.com (J.R.I.); marcelafrancoti@gmail.com (M.F.T.); barbara.brum@ufu.br (B.d.A.B.); daise.rossi@ufu.br (D.A.R.)

**Keywords:** antimicrobial resistance, chilled meat, frozen meat, RAPD, virulence

## Abstract

We aimed to identify the prevalence of thermophilic species of *Campylobacter* in meats of different species available on the Brazilian commercial market and to determine the genetic diversity, antimicrobial resistance and virulence potential of the isolates. A total of 906 samples, including chicken, beef and pork carcasses and chicken and beef livers, were purchased in retail outlets, and prevalences of 18.7% (46/246), 3.62% (5/138), 10.14% (14/138), 3.62% (5/138) and 4.47% (11/132), respectively, were identified, evidencing the dissemination of genotypes in the main producing macro-regions. Of all isolates, 62.8% were classified as multidrug resistant (MDR), with resistance to amoxicillin-clavulanate (49.4%), tetracycline (51.8%) and ciprofloxacin (50.6%) and co-resistance to macrolides and fluoroquinolones (37.1%). Multivirulent profiles were identified mainly in isolates from chicken carcasses (84.8%), and the emergence of MDR/virulent strains was determined in pork isolates. All isolates except those from chicken carcasses showed a high potential for biofilm formation (71.4% *luxS*) and consequent persistence in industrial food processing. For chicken carcasses, the general virulence was higher in *C. jejuni* (54.3%), followed by *C. coli* (24%) and *Campylobacter* spp. (21.7%), and in the other meat matrices, *Campylobacter* spp. showed a higher prevalence of virulence (57.2%). The high rates of resistance and virulence reinforce the existence of strain selection pressure in the country, in addition to the potential risk of strains isolated not only from chicken carcasses, but also from other meat matrices.

## 1. Introduction

*Campylobacter* spp. are the most prevalent bacterial etiologic agents in foodborne gastroenteritis in developed countries and were officially responsible for 59.7 cases of campylobacteriosis per 100,000 of the population in 2020 [1,2]. Chicken meat has been the main source of human infection since 2008, but the consumption of meat from other animals also participates in the chain of transmission, since the species that make up the genus are capable of colonizing different farm animals [2,3].

Although campylobacteriosis affects more than 400–500 million people worldwide [4], and Brazil is an important producer and exporter of animal protein worldwide, there is no official record in the country of cases of human Campylobacteriosis in the last 10 years, and there are no regulations for the systematic or regulatory monitoring of the pathogen in meat sold directly to the population. Thus, even with records of the presence of *Campylobacter* in different products of animal origin in research in other countries [5], these reports are sporadic and cover specific regions and conditions in Brazil, which encourages the search for real knowledge about the presence, number, virulence potential and dissemination of this bacterium in meat from different species, as well as the determining factors of its presence, including the type of sanitary inspection and how the product is marketed, frozen or chilled.

The pathogenesis of *Campylobacter* infection is complex and not yet fully elucidated, but like other pathogens, in addition to the host–parasite relationship, it possesses the gene apparatus to colonize, invade, produce cytotoxins, and perpetuate itself in the environment [6]. This includes different genes involved in colonization and adhesion (*pdlA* and *cadF*), cell invasion (*ciaB*), motility (*flaA*), stress adaptation (*dnaJ*), toxin production (*cdtA, cdtB* and *cdtC*) and quorum sensing communication (*luxS*), which play a significant role in disease development. In addition, being resistant to different classes of antimicrobials demonstrates, along with their prevalence, an increased risk of infecting hosts and causing conditions that are more severe or difficult to treat [7,8].

Considering the prominent position of the Brazilian meat industry in the global market, Brazil’s status as the world’s largest exporter of chicken meat [9], the underreporting in the country regarding contamination and infection by *Campylobacter* and the scarcity of studies that characterize this pathogen at the epidemiological, molecular and phenotypic levels, we propose a panoramic analysis of the occurrence of *Campylobacter* in meats marketed in Brazil. We included in the investigation the prevalence of the genus, the main species of public health importance, antimicrobial resistance, virulence potential and genetic diversity.

## 2. Materials and Methods

### 2.1. Sampling and Microbiological Analysis

We analyzed 906 meat samples from different species, representative of 53 Brazilian commercial brands, during the period from August 2014 to February 2016, under different types of sanitary inspection (Federal, State and Municipal), qualified for domestic trade (State and Municipal-internal consumer market) and/or export (Federal-external consumer market). Samples were obtained from a total of 31 producers, of which 14 were representative brands of the main chicken carcass markets in the country, collected directly from the commercializing markets, under coordination of the National Health Surveillance Agency (ANVISA). The number of samples analyzed to determine prevalence was established according to Thrusfield [10]:

Table 1 shows how the number of samples, sampling unit and isolation protocol for each type of matrix were defined.

Protocol A: for chicken carcasses. The sample aliquot was obtained using the rinsing technique, with half of the carcass placed individually in a sterile plastic bag containing 225 mL of sterile 0.1% peptone water (Difco^®^). The samples were subjected to a process of agitation and massaging for 60 s, with the most vigorous massaging on the neck, armpit, chest and groin. The product of the lavage was quantitatively and qualitatively analyzed for *Campylobacter* (Step 1). The other half of the carcass was submitted to freezing in a domestic freezer (−18 to −20 °C) for 30 days for later evaluation of survival and the number of viable *Campylobacter* cells for samples that were positive in the first stage (Stage 2).

Protocol B: for pork and beef liver samples. The methodology consisted of weighing 150 g of each meat matrix in 100 mL of sterile 0.1% peptone water (Difco^®^). The samples were submitted to the same rinsing technique as described in Protocol A, and the rinse water was used for counting and analysis of the presence/absence of *Campylobacter* (Step 1). Step 2 consisted of storing another 150 g of each matrix and evaluating the same parameters after 30 days under freezing in the samples that were positive in the first step.

Protocol C: intended for samples of ground beef and chicken liver. We used 10 g of the matrix diluted in 90 mL of Bolton broth (Oxoid^®^). The samples were homogenized and analyzed quantitatively and qualitatively for *Campylobacter* (Step 1). The rest of the matrices were kept at –20 °C for 30 days for analysis of the samples that were positive in the first step (Step 2).

Thirty milliliters of the rinsate was added to 30 mL of doubly concentrated Bolton broth (CM0983, Oxoid, Hampshire, UK) supplemented with an antibiotic mix (SR0183E, Oxoid) and with 5% horse blood (Laborclin, Paraná, Brazil) (Protocols A and B). Protocol C samples were incubated directly (10 g in 90 mL Bolton broth supplemented with 5% defibrinated sheep blood and an antibiotic mixture). Before incubation, 1-mL aliquots of the samples diluted in Bolton broth (protocol C) or in 0.1% peptone water (protocols A and B) were used to perform serial decimal dilutions and quantification of *Campylobacter* in *Campylobacter* Blood-Free Selective Agar (Modified CCDA-Preston) (CM0739, Oxoid). Bolton broth tubes were incubated in a microaerobic atmosphere (5–15% O_2_ and 10% CO_2_) using a Microaerobac (Probac do Brasil, Sao Paulo, Brazil) at 37 °C for 44 ± 4 h. After Bolton broth incubation, a membrane filtration method was used to plate the samples on *Campylobacter* Blood-Free Selective Agar (Modified CCDA-Preston) (CM0739, Oxoid) plates supplemented with an antibiotic (SR0155E, Oxoid). Briefly, a 0.65-µm-pore-size cellulose membrane filter (Millipore, MA, USA) was placed on top of the medium, and 300 µL of each enrichment in Bolton broth was added to the plate. After approximately 15 min, the membrane was dry, and it was removed from the agar plate. Modified CCDA-Preston plates were incubated at 37 °C for 44 ± 4 h in a microaerobic atmosphere, as described above.

For quantification, 1-mL aliquots of the samples diluted in Bolton broth or 0.1% peptone water (protocols 1, 2 and 3) were used to perform serial decimal dilutions in 9 mL of 0.1% peptone water (DifcoTM). After this procedure, 100 µL of the respective dilutions were inoculated in CCDA agar plates, supplemented with antibiotics and covered with a 0.65-µm-pore-size cellulose membrane. After removal of the membrane, samples were incubated at 37 °C for 44 ± 4 h in a microaerobic atmosphere, followed by calculations for the expression of quantitative results.

### 2.2. Molecular Analysis of Specific Genes, Transcript Production and Genetic Similarity

After isolation, two colonies typical of *Campylobacter* spp. were randomly selected from each modified CDDA-Preston plate for further analysis. The selected colonies were resuspended in Bolton broth (Oxoid) and cultured overnight. Total DNA was extracted using a commercial kit (Wizard Genomic DNA Purification kit, Promega, Madison, WI, USA), according to the manufacturer’s instructions. All PCR reactions performed for species identification, virulence genes, virulence transcripts and similarity analysis are described in Table 2. 

The ability to produce transcripts for *ciaB* (invasion), *dnaJ* (thermotolerance), *p19* (iron uptake) and *sodB* (protection against oxidative stress) was investigated by qualitative reverse transcription PCR (RT-PCR) [14,15]. RNA extraction was performed using the Trizol method according to Li et al. [15]. The RNA concentration used was 200 ng/μL, quantified in a NanoDrop spectrophotometer (Thermo Scientific^®^, Wilmington, DE, USA).

Reverse transcription was performed with 10 U of RNase inhibitor, 40 U of MMLV-RT (Amersham Biosciences^®^, Saint Louis, MO, USA), 1X MMLV-RT buffer (Amersham Biosciences^®^), 200 μM of dNTPs (dGTP, dATP, dTTP and dCTP), 126 pmol of random hexamer oligonucleotides as primers (Invitrogen, Carlsbad, CA, USA), 20 μL of DEPC water (Invitrogen^®^) and 1 μL of RNA, all maintained at 37 °C for one hour to obtain complementary DNA (cDNA). Subsequently, 3 μL of cDNA was used for amplification in a 25-μL reaction volume, comprising 0.625 U of Taq DNA polymerase, 5 mM MgCl_2_, 200 μM of dNTPs and 4 pmol of each primer (Table 2) (Invitrogen). 

The genetic diversity among the isolates was determined by the RAPD-PCR (Random Amplification of Polymorphic DNA) technique, according to the protocol described in Table 2. 

At the end of the reactions, the amplified products were submitted to electrophoresis in 1.5% agarose gel, stained with SYBR safe solution (Invitrogen Brasil Ltd), submitted to a voltage of approximately 8 V/cm and subsequently visualized with UV light in a transilluminator (L.PIX Loccus Biotechnology, Cotia, Brazil).

The RAPD-PCR results were evaluated using the GelCompar II program (Comparative Analysis of Electrophoresis Patterns, version 1.50, Applied Maths Korthrijk, Sint-Martens-Latem, Belgium) with the Dice similarity coefficient, with 1% tolerance for each primer separately. For dendrogram construction, the UPGMA method (unweighted pair group method with arithmetic mean) used the average of all experiments. 

### 2.3. Susceptibility to Antimicrobials 

Resistance to antibiotics in *Campylobacter* was assessed by the disk diffusion method against six antibiotics routinely used to treat infections in humans and in veterinary medicine. Samples were suspended in 0.9% NaCl to obtain 5 × 10^5^ CFU/mL (0.5 McFarland Standard) and seeded on Mueller Hinton agar plates supplemented with 5% defibrinated sheep blood (Laborclin). The following antimicrobials were tested: amoxicillin and clavulanic acid (10 µg), azithromycin (15 µg), ciprofloxacin (5 µg), erythromycin (15 µg), gentamycin (10 µg) and tetracycline (30 µg) (Oxoid^®^), according to EUCAST [25]. Plates were sealed and incubated at 41 ± 1 °C for 40–48 h in microaerobic conditions as described above. Subsequently, the strains were classified as sensitive (S) or re-susceptible (R) to the antimicrobial tested *C. jejuni* IAL 2383 and *C. jejuni* NCTC 11351 were used as positive controls, whereas a blank sample was used as a negative control. For those antimicrobials not classified for *Campylobacter*, the de-defined standard for *Enterobacterales* was used.

### 2.4. Statistical Analysis

To compare rates between groups, the binomial test for two proportions was used at 5% significance (Action Tool (2015) using the R program (R Development Core Team, 2015). The results were used to estimate the public health risk posed by the consumption of the chicken carcasses. Fisher’s exact test was used to compare *Campylobacter* positivity between the different matrices, forms of preservation (frozen/chilled), inspection systems, commercial brands and years of isolation. The calculations were performed in the GraphPad-Prism 8.0 program.

## 3. Results

### 3.1. Prevalence of Campylobacter on Chicken Carcasses and in Meat Matrices

*Campylobacter* spp. were identified in a total of 18.69% (46/246) of the chicken carcass samples and 5.3% (35/660) of the different types of meat evaluated, including chicken liver and chilled and frozen pork and beef (Table 3).

The prevalence of the different *Campylobacter* species in chicken carcasses and in the other meat matrices were, respectively, *C. jejuni* (54.3%, 25/46, and 17.1%, 6/35), *C. coli* (23.9%, 11/46, and 25.7%, 9/35) and the other *Campylobacter* spp. (21.7%, 10/46, and 57.1%, 20/35). *C. jejuni* was the most common species in chicken carcasses (*p* = 0.001, Fisher test) and *Campylobacter* spp. in the other breeds (*p* = 0.002, Fisher test). The prevalence of *Campylobacter* in chicken liver (3.62%) and beef liver (4.48%) was lower than expected.

All positive samples were confirmed by the quantitative method, with a limit of quantification of 10 CFU/g. In 8/81 (9.9%) samples (6 from chicken carcass and 2 from beef liver), the enumeration was accurate and equivalent to an average of 123 CFU/g (minimum = 10 and maximum = 468 CFU/g). For the other samples, there was confluent growth indicating counts higher than 10^7^ CFU/g [26], which determined statistical equivalence in the levels of contamination by *Campylobacter* spp. in the different meat matrices (Kruskal–Wallis test, *p* = 0.352).

Freezing did not represent a form of *Campylobacter* spp. control, since the prevalence in both forms of commercial preservation was statistically equal (*p* = 0.412, Fisher test). However, slow (in-home) subsequent freezing of the positive samples did not maintain *Campylobacter* viability, since in the second stage (samples kept frozen for 30 days after processing, described in the methodology: stage/step 2), we reisolated the bacteria only in 2/46 chicken carcass samples originally marketed frozen (one *C. jejuni* and one *Campylobacter* spp.).

The type of inspection system did not influence the prevalence rates of *Campylobacter* (*p* > 0.05), but the positivity rate was lower (*p* = 0.023) in companies that export their products (28/184; 15.22%) than in those that sell to the domestic market (18/62; 29.03%). We obtained a significantly higher prevalence in the Southeast (35/144; 24.31%) than in the Midwest (10/92; 10.81%; *p* = 0.011) and South (1/10; 10%; *p* = 0.045) regions (Figure 1).

### 3.2. Antimicrobial Resistance Phenotypes

The least effective antimicrobials for controlling the isolated strains were tetracycline (51.85%) and ciprofloxacin (50.62%). We found a significantly higher percentage of resistance to tetracycline among *Campylobacter* spp. isolates. (66.7%) than among *C. coli* (35%) (*p* = 0.043, Fisher test). For ciprofloxacin, high resistance was attributed to strains isolated from chicken carcasses (60.9%, *p* = 0.045 Fisher test). The highest susceptibility was identified for the antimicrobial’s azithromycin and gentamicin, with 72.84% and 67.9%, respectively (Table 4).

Analysis of resistance profiles identified 35 types (Table 5), of which those with multidrug resistance (three or more classes of unrelated antimicrobials) were the most common (22/35 profiles, 62.8%, and 44/81 strains, 54.3%), statistically different from the number of susceptible, mono- and co-resistant strains (*p* < 0.0001). The number of MDR strains did not differ (*p* < 0.05) according to species: 18/30 (60.0%), 18/31 (58.1%) and 8/20 (40.0%) of the isolates corresponded to *Campylobacter* spp., *C. jejuni* and *C. coli*, respectively. The P35 profile, which includes resistance to all antimicrobials tested, was representative of isolates belonging to the other meat sources, except for chicken carcasses (*p* = 0.032). Moreover, in the other meat matrices, we detected higher MDR in *C. jejuni* species versus *C. coli* (*p* = 0.031). Co-resistance to macrolides and fluoroquinolones was identified in 13/35 (37.1%) profiles, which included 26/81 (32.1%) strains and did not differ according to the matrix type or species (*p* < 0.05).

### 3.3. Characterization of Virulence Factors

All strains presented at least one of the virulence genes studied, indicating that they have different degrees of potential virulence. The most identified genes were *cadF* (62/81), *ciaB* (50/81), *pldA* (49/81) and *flaA* (44/81), and the least identified were *sodB* and *cdtABC* (Table 6).

The prevalence of 8/10 virulence genes were evident in samples from broiler carcasses compared to isolates from the other matrices (*p* < 0.02, Fisher test). The exceptions were the *sodB* gene, poorly identified in the samples (2/81, 2.5%), and the *luxS* gene, more prevalent in isolates from other meat sources (25/35, 71.4%; *p* = 0.001, Fisher test). *C. jejuni* showed significantly higher virulence percentages for the *ciaB* (*p* = 0.042), *cdtABC* (*p* = 0.018) and *luxS* (*p* = 0.009) genes compared to the other species, and *C. coli* showed a high prevalence of *pldA* (*p* = 0.034), *flaA* (*p* = 0.002) and *cadF* (*p* = 0.021).

Discrimination into profiles allowed the identification of 50 variations in the virulence characterization of the strains, of which 30 profiles were of isolates exclusively from chicken carcasses, and 19 were from other meats. Only one profile contained one strain from pork shank and one from chicken carcass. We observed that chicken carcasses represented the main source of virulent *Campylobacter* (*p* < 0.0001, Fisher test), with 39/46 (84.8%) showing four or more virulence genes, in contrast to the other matrices (9/35; 25.7%). It is worth noting the emergence of virulent strains in the pork shank matrix, which, besides presenting four of the nine strains with four or more virulence genes, showed greater potential for invasion and biofilm formation (*p* = 0.042), identified by the presence of high percentages for genes *ciaB* (9/14, 64.3%), *pldA* (6/14, 42.8%), *cadF* (9/14, 64.3%) and *luxS* (12/14, 85.7%). *C. jejuni* (*p* = 0.001) and *C. coli* (*p* = 0.042) showed higher virulence in relation to *Campylobacter* spp., and *C. jejuni* grouped the eight profiles with more virulence genes (V43-50) (Appendix A). We also observed that strains from frozen samples were more likely to be more virulent (27/39, 69.2%) than those from chilled samples (14/42, 33.3%) (*p* = 0.006, OR = 4.0, Fisher test).

The analysis of virulence transcript production was restricted to the 46 strains obtained from chicken carcasses. It is worth noting that 17/46 (36.9%) of the strains did not transcribe any of the genes tested. Transcription of the *dnaJ* gene (23/46; 50%) was identified in *C. jejuni* (48%), *C. coli* (63%) and *Campylobacter* spp. (40%), with no statistical difference among species (*p* > 0.05). The production of transcripts of the *sodB* gene was observed in only 4.3% of isolates, all from *C. jejuni*. The *p19* and *ciaB* genes were transcribed in 50% and 30.4% of strains, respectively, with no statistical difference among species (*p* < 0.05). We observed that for *C. jejuni*, the gene transcription process was more evident, since the joint transcription of all or three of the evaluated genes (except *sodB*) was exclusive to this species (6/46; 13.0%). The condition of the sample (chilled/frozen) did not interfere in the transcription of the genes studied (*p* > 0.05).

### 3.4. Genetic Similarity

In the evaluation of genetic similarity in chicken carcasses, three dendrograms were discriminated for 40/46 strains, with one for each species: *Campylobacter* spp. (Figure 2a), *C. jejuni* (Figure 2b) and *C. coli* (Figure 2c), and no clones were present. The strains that lacked RAPD profiling included two strains of each species (two *C. jejuni*, two *C. coli* and two *Campylobacter* spp.).

The strains of *Campylobacter* spp. were no more than 80% similar, indicating the probable presence of several other *Campylobacter* species (other than *C. jejuni* and *C. coli*). All were unique to the year 2015 in the MG state and with the common presence of the *cadF*, *ciaB* and *pldA* genes and resistance to ciprofloxacin and tetracycline. Three clusters were identified in *C. jejuni* (profiles of A, B and C). Cluster A grouped two strains with a similarity of 90.1% from frozen samples from Minas Gerais state in August and September 2015, which had in common the *ciaB*, *cadF* and *pldA* genes and resistance to GEN and ERY. Eight strains included cluster B, with 85.9% homology, from different origins (MG, MT and GO), from chilled and frozen samples produced in the years 2014 and 2015, under different inspection systems and distinct virulence and antimicrobial resistance profiles. The five strains that made up group C had in common the fact that they came from frozen chicken carcass samples, along with the presence of the *flaA*, *ciaB*, *cdt*, *cadF* and *pldA* genes and resistance to ciprofloxacin. 

*C. coli* strains were grouped into two clusters. For cluster D, three strains from different regions were grouped, which had in common the presence of the *flaA*, *pldA* and *cadF* genes and resistance to ciprofloxacin. Cluster E included four strains with the presence of the *cadF* gene in common.

For the other meat matrices, the dendrograms are shown in Figure 3 and included 32/35 strains that were discriminated into four clusters for *Campylobacter* spp. (A, B, C and D), three for *C. coli* (E, F and G) and two for *C. jejuni* (H and I). The strains that did not show a RAPD profile included one from chicken liver (*Campylobacter* spp.) and two from pork shank (*C. coli* and *Campylobacter* spp.).

In *Campylobacter* spp. (Figure 3a), the distribution of strains from distinct meat matrices within the same RAPD genotype was common. Especially, in cluster B, we detected a 92% similarity between samples originating from pork shank and chicken liver in the year 2015, with the presence of the *luxS* gene in common. 

For *C. coli* (Figure 3b) and *C. jejuni* (Figure 3c), we observed that the clusters were discriminated according to the meat matrix, annual production and industrial and sanitary inspection services. In *C. coli*, the presence of the *flaA* and *cadF* genes was common to clusters E and F. In cluster E, two chicken liver strains were grouped together, with 99.2% similarity, from samples stored under refrigeration and freezing, from brands produced in different states (MT and GO), indicating the dissemination of the genotype among different establishments. For *C. jejuni*, both clusters (H and I) contained the *ciaB* and *luxS* genes. Cluster H, composed of three strains from pork shank and one from chicken liver, showed a similarity of 81.8%, demonstrating the presence of common genotypes circulating in different production chains.

## 4. Discussion

### 4.1. Campylobacter Positivity 

The highest prevalences of the *Campylobacter* genus were found in chicken carcasses and pork shank and corroborated other recent studies in carcasses and in slaughterhouses [27,28]. However, the prevalence of *Campylobacter* in liver meat varies widely from country to country and can reach extremely high rates (96%) [29,30]. The lowest values found for beef liver and chicken liver in our study were not expected, since this kind of meat is a maintenance niche for bacteria and a source of foodborne outbreaks [31]. Thus, it is possible that the bacterial injury condition potentiated the acquisition of the viable but nonculturable form, reducing our prevalence values.

*C. jejuni* was the most frequently isolated pathogen in poultry carcasses, and in meat matrices, *Campylobacter* spp. were the most prevalent, which are justified by differences in contamination in the pre-slaughter sector, reservoir animals, retail meat origin and regions of origin [32].

Despite the presence of samples with low counts (<1000 CFU/g), the others showed confluent growth indicative of counts higher than 10^7^ CFU/g [26]. According to EU legislation, the maximum count allowed in up to 60% of sampling is 1000 CFU/g. The confluent growth pattern identified in the other samples raises concerns about the level of contamination, providing high susceptibility to infection [33].

The cold chain has been indicated as a way to decrease or control pathogen load in food [34], but freezing of commercial samples was not an effective means of controlling bacteria. Research with *C. jejuni* in chicken meat demonstrated that the combination of refrigeration and freezing were no substitute for safe handling and proper cooking of poultry [35]. A study by MAPA (2021) demonstrated that the time between product manufacture and laboratory analysis can be decisive in the number of samples showing high counts for *Campylobacter* spp. [36], but in our study, this factor was not relevant since there was no difference in this period between the negative (mean of 42.69 days) and positive frozen samples (mean of 50.36 days). This allows us to infer that some strains with a greater ability to adapt may be selected under adverse conditions and that freezing does not guarantee a food free of *Campylobacter* spp.

In parallel, our study found that subsequent slow freezing of the samples for 30 days allowed the recovery of only 2/46 (4.3%), probably due to the long duration of injury and acquisition of the viable but nonculturable form.

Exporting companies presented a lower rate of *Campylobacter* isolation, probably due to the higher stringency of legislation to meet the foreign market, which ensures greater biosecurity in production activities. The main strategies for pathogen control in the food export industry include the application of specific food safety protocols, such as the Exploratory Program for Research and Estimation of Prevalence of *Campylobacter* spp., in conjunction with the monitoring and control of *Salmonella* spp., instituted by the MAPA normative instruction, determined by IN No. 20, of 21 October 2016 [36] and programs for strict hygienic practices (HACCP), which focus on the reduction or absence of the pathogen in the final product [37].

Southeastern Brazil is the country’s second largest chicken-producing region, which justifies the highest percentage of isolation in this location, which includes the states of Minas Gerais and São Paulo [38].

### 4.2. Resistance of Campylobacter Isolates

Resistance to AMC presented with a relatively high rate and was aggravated by the fact that it is a β-lactam widely used in animal treatment, with several registrations and commercial brands approved by the Ministry of Agriculture (MAPA) [39]. Resistance to AMC has been associated with the production of β-lactamases, and its exacerbated use may contribute to the selection of carbapenem-resistant strains [40]. It is known that Brazil is among the 10 largest consumers of veterinary antimicrobials recognized worldwide and lacks complete reports regarding their use [41], which determines the need for more rigorous control.

The low efficacy of TET is impactful from a clinical point of view, since it is an alternative drug that can be administered in the treatment of campylobacteriosis [42]. In addition, TET has been banned in the country as a growth promoter since the year 2009 [43], but it was used extensively in the past, which may have selected resistant strains that have remained circulating.

For CIP, we also identified low efficacy, which was mainly attributed to chicken carcass strains. The cause of quinolone resistance in the poultry chain is its overuse in production, treatment and disease control, in addition to the spread of resistance in human strains as an aggravating factor [44].

Co-resistance to macrolides and fluoroquinolones in 37.1% of strains raises concerns about treatment for campylobacteriosis, since both represent drugs of choice [45] and are considered critically important antibiotics [46]. The concern is amplified for quinolones, considering their maintenance in South American wastewater due to uncontrolled veterinary practices [47]. For macrolides, the implementation of stringent control measures [48] may have determined the higher susceptibility of our strains to AZI and ERY.

GEN also showed good efficiency, which contributes to the institution of this drug in the therapy in severe cases of campylobacteriosis [49], especially when there is resistance to the drugs of choice (macrolides and fluoroquinolones) [50].

MDR profiles were more prevalent, which intensifies concern about the treatment of campylobacteriosis. The prevalence of MDR strains varies in the literature [51,52]. *C. jejuni* was the most common MDR species in samples of other meat matrices. Furthermore, resistance to all the tested drugs was identified exclusively in swine and bovine meat matrices. These facts point to the consequences of intensive antibiotic application. which promotes intense selection pressure on MDR *Campylobacter* at the end of slaughter processing, aggravating the problem at the interface of food consumption and human disease [53].

### 4.3. Virulence of Campylobacter Isolates

The variability in the virulence potential identified by the diversity of virulence profiles (50) reflects the genetic plasticity of *Campylobacter*, which represents a genus in an intense process of evolution. The high occurrence of recombination events generates alterations in its genetic material that facilitate gene flow among species of the same genus and drive its high diversity [54].

The higher frequency of *cadF*, *ciaB*, *pldA* and *flaA* in the strains reflects their virulence potential. The *cadF* gene plays a primary role in adhesion, which represents a prerequisite for colonization of the host gastrointestinal tract. The ability of *Campylobacter* to adhere to fibronectin is mediated by adhesins on the surface of the bacterium encoded by this gene [55]. Cell invasion and intestinal colonization are aided by expression of the *pldA* gene [19]. Furthermore, in animals, the presence of strains positive for this gene assists in the maintenance of commensal relationships [56]. Rivera Amil et al. [57] described that *ciaB* acts by encoding a protein that functions to destroy microtubules and thus facilitates cell invasion and infection in the host. The *flaA* gene is highly conserved in *Campylobacter* and is important for survival under the internal conditions of the gastrointestinal environment [58]. The absence of this gene may reduce colonization ability and alter *Campylobacter* motility, but it also allows for greater antigenic variation, which represents a strategy for evading the host immune system [59]. 

The low frequency of *sodB* indicates that the strains are less able to maintain viability under oxidative stress conditions. This gene encodes the production of SOD (superoxide dismutase) proteins that assist in the survival of *Campylobacter* under stress conditions. Under heat stress (chilling/freezing), there is an intense production of free radicals by the bacterial cell, which, in the absence of SOD, induces the process of injury and intracellular dehydration [60]. 

The identification of the *dnaJ* gene in 53% of strains determines the potential to encode heat-shock-related proteins, which allows for tolerance to sudden temperature variations [61].

The cytolethal distending toxin (CDT) is encoded by three adjacent genes, *cdtA, cdtB* and *cdtC*, and is involved in cell cycle arrest in the G2/M phase [62]. This cytotoxin induces a cytoplasmic distension, leading to cell death by apoptosis about three to four days after infection, contributing to the development and pathogenesis of inflammatory diarrhea caused by campylobacteriosis in humans [63]. The low percentage of *cdtABC* found in our study (<31%) was not expected [64] but is similar to the Angelovski et al. [65] study. 

The greatest potential for biofilm formation (*luxS* gene) was detected in strains from other meat matrices. This gene is the most important in the acquisition of the fundamental sessile form in cell–cell communication (quorum-sensing; QS), in the structuring of biomass and in the recognition and inclusion of bacterial populations [66]. Thus, the higher potential for biofilm formation in strains from the other meat matrices also constitutes one of the main strategies of *Campylobacter* to resist multiple stress conditions, including low temperatures and competitive exclusion caused by the more prevalent cohabitant microbiota in these matrices. In addition, generalist isolates of *C. jejuni* possess the ability to survive for long periods under aerobic conditions within the microenvironments present in biofilms, which contributes to their persistence in primary animal protein processing [67].

Strains from chicken carcasses showed a higher virulence potential, which is in agreement with the literature [68], which reinforces that these strains may be under the influence of selection pressure imposed by the conditions maintained in an industrial environment [69]. It is worth mentioning that specifically for pork shank, there is an emergence of virulent strains, which is a warning, since few studies have shown this profile [28]. 

Strains from frozen samples presented a four-times greater chance of being highly virulent (presence of four or more virulence genes). This fact is alarming, since it indicates that freezing can determine a selection pressure for more virulent strains, since stress conditions impact the virulence and pathogenesis of bacterial strains [70]. 

The similarity in virulence of *C. jejuni* and *C. coli* shows that although many papers indicate *C. jejuni* as the more virulent species, it is possible that *C. coli* has the same potential acquired through cohabitation with *C. jejuni*. Inter-species recombination, especially between *C. jejuni* and *C. coli*, plays an important role in the evolution of the genus *Campylobacter*. In fact, about 18.6% of the allelic variants identified in *C. coli* exhibit ancestry from *C. jejuni*, while only 2.3% of *C. jejuni* alleles were acquired from *C. coli*, indicating an asymmetric gene flow between the two species and justifying the increased incidence of *C. coli*-related cases of campylobacteriosis [71].

The discrimination of unique profiles of chicken carcasses and the other meats reflects specific diversities according to origin. *Campylobacter* is known to exhibit a multi-host lifestyle that reveals its broad genetic diversity. Genetic sequencing analyses have shown the specificity of certain profiles harboring typical origins, termed host specialists, whose presence of an interaction barrier can be identified by the separation of host niches, such as those identified in chicken carcasses in our study. Other strains, called host generalists, may colonize a wide range of animal reservoirs, similar to what we observed for virulence profiles unique to other meat matrices [72].

The injury condition (chilling/freezing) of the isolates represents an important factor in the modulation of the transcription process and can explain the absence of transcripts in 17/46 isolates and the high transcription of stress-related genes (*dnaJ* and *p19*). 

The lower transcription of *ciaB* is justified by the modulation of transcription according to the needs of *Campylobacter*, with *ciaB* being more expressed when the bacterium finds a favorable invasion condition inside the host [73]. The low transcription of *sodB* was expected, since only two strains presented the gene and both produced transcripts. The production of transcripts identified exclusively in *C. jejuni* for at least three of the investigated genes indicates its greater adaptation to adverse conditions (cooling/freezing), which allows better regulation and maintenance of the transcription process.

### 4.4. Similarity of Campylobacter Isolates

The identification of bands by RAPD analysis was not possible in all strains, due to failure of DNA amplification indicated by the absence of banding or the possible interference of nucleases; both cases can be identified in several strains [74]. 

For *Campylobacter* spp. from chicken carcasses, the failure to cluster reflects the presence of distinct species common in some regions, which include simultaneous contamination with *C. consius*, *C. foetus*, *C. hyoilei*, *C. lari* and *C. mucosalis* [75].

The formation of clusters containing strains from two or more meat matrices (Figure 3a) in *Campylobacter* spp. indicates greater proximity between isolates from distinct matrices. Modeling studies on genomic plasticity in *Campylobacter* demonstrate that this pathogen presents a generalist gradient that allows colonization of new niches and optimizes the ability to survive rapid transitions in different hosts, evidencing that the acquisition of the same strain may originate from more than one source [54]. 

Persistence of strains in the same state and in different periods has been identified in *C. jejuni* isolated from chicken carcasses. This maintenance indicates disregard of biosecurity standards and self-control programs coupled with the potential of the strains to establish themselves in the sessile life form and determine recurrent contamination of the final product. Indications of reintroduction and persistence of genetically identical strains have been investigated during the production cycle in broiler and turkeys, and intervention measures such as empty periods and microbial decontamination techniques reduce *Campylobacter* spp. prevalence in primary processing [76]. 

The spread of genotypes across different regions of the country was the most identified fact in the strain similarity analysis. This indicates that the perpetuation of similar strains in different regions occurs due to the trade of products and that they can determine the dissemination of these profiles at a global level, considering the importance of the country in the export of animal protein [77].

## 5. Conclusions

We conclude from this study that multiple factors are involved in *Campylobacter* epidemiology, allowing a distribution among regions and the persistence of strains in chilled and frozen meats of different origins. The higher prevalences in chicken carcasses indicate the main focus of control in the country, but monitoring in bovine and porcine carcasses should be a central point to extend control. *C. jejuni* is the most important species in the country, but monitoring should be extended to other *Campylobacter* species, considering the high levels of resistance and virulence identified and the genetic plasticity of this genus. Selection pressure for emerging freeze-adapted, multivirulent and MDR strains indicates the need for a comprehensive assessment of *Campylobacter* molecular mechanisms as a measure to support the implementation of widespread surveillance strategies for this pathogen.

## Figures and Tables

**Figure 1 ijerph-19-06087-f001:**
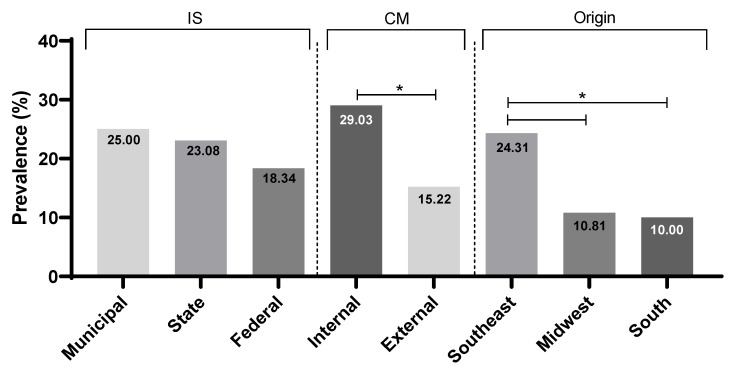
Prevalence of *Campylobacter* spp. in chicken carcasses according to the inspection system, * *p* < 0.05, Fisher’s exact test.

**Figure 2 ijerph-19-06087-f002:**
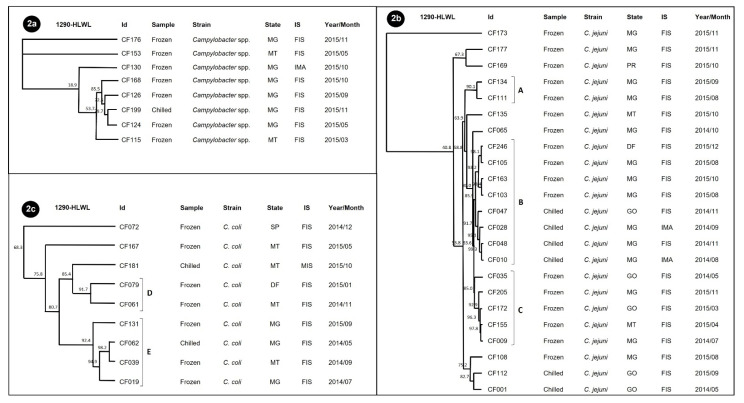
Dendrogram of the 40 *Campylobacter* spp. isolates from chilled and frozen chicken carcasses, from data generated by the RAPD-PCR technique with primers 1290 and HLWL, using the average from experiments and the UPGMA method with 85% optimization by the GelCompar program. (**2a**) Results found for *Campylobacter* spp. (**2b**) Results found for *C. jejuni*: (**A**) cluster with 90.1% homology; (**B**) cluster with 85.9% homology; (**C**) cluster with 85% homology. (**2c**) Results found for *C. coli*: (**D**) cluster with 91.7% homology; (**E**) cluster with 92.4% homology. SP (São Paulo), MT (Mato Grosso), MG (Minas Gerais).

**Figure 3 ijerph-19-06087-f003:**
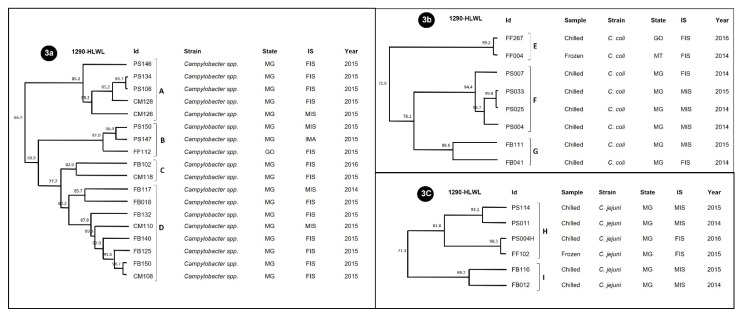
Comparative dendrogram of 32 *Campylobacter* strains for other meat matrices using the Dice similarity coefficient with 1% tolerance and the UPGMA method with 0.80% optimization. (**3a**) Analysis of *Campylobacter* spp.; (**3b**) Analysis of *C. coli*; (**3c**) Analysis of *C. jejuni*. (A–I) clusters formed with homology greater than 80%. CM—ground bovine duckling; FB—bovine liver; FF—chicken liver; PS—pork shank. SP (São Paulo), MT (Mato Grosso) and MG (Minas Gerais).

**Table 1 ijerph-19-06087-t001:** Number, sample types and sampling methods used in the study.

Meat Matrix	*Pexp **	Chilled	Frozen	Total	Isolation Protocol ISO 10272-1:2006 (ISO, 2006)	Matrix Portion
Chicken carcasses	80% ^1^	80	166	246	A: Rinsing	⅟_2_ carcass
Pork shank	10% ^2^	138		138	B: Rinsing	150 g
Bovine liver	80% ^3^	132	114	246	B: Rinsing	150 g
Chicken liver	90% ^4^	24	114	138	C: Homogenization	10 g
Minced meat	10% ^2^	138		138	C: Homogenization	10 g
**Total**		512	394	906		

*Pexp*: expected prevalence; * 95% confidence level (Thrusfield, 2004) [10]; ^1^ (FAO, 2009) [11]; ^2^ (Zhao et al., 2010) [5]; ^3^ (Noormohamed; Fakhr, 2013) [12]; ^4^ (Whyte et al., 2006) [13].

**Table 2 ijerph-19-06087-t002:** *Primers*, function, amplicon size, PCR conditions, RT-PCR, RAPD-PCR and references.

Genes	Function	*Primers*	Sequence 5′ → 3′	Size (bp)	DNA (ng)	*Primer* (pmol)	PCR Condition	Reference
** *16S-rRNA* **	Gender identification	16S-rRNA-F 16S-rRNA-R	ATCTAATGGCTTAACCATTAAAC GGACGGTAACTAGTTTAGTATT	857	30	40	94 °C—1 min; 25 cycles: 94 °C—1 min, 60 °C—1 min, 72 °C—1 min; 72 °C—7 min	Linton et al. [16]
** *pg* **	Multiplex PCR: Identification of *C. jejuni* and *C. coli*	pg3 pg50	GAACTTGAACCGATTTG ATGGGATTTCGTATTAAC	460	20	40	94 °C—4 min; 25 cycles: 94 °C—1 min, 47 °C—1 min, 72 °C—1 min; 72 °C—7 min	Harmon et al. [17]
** *C* **		C1 C4	CAAATAAAGTTAGAGGTAGAATGT GGATAAGCACTAGCTAGCTGAT	160		20		
** *flaA* **	Motility	flaA-F flA-R	ATGGGATTTCGTATTAACAC CTGTAGTAATCTTAAAACATTTTG	1728	20	30	95 °C—10 min; 35 cycles: 95 °C—1 min, 45 °C—1 min, 72 °C—2 min; 72 °C—10 min	Hänel et al. [18]
** *pdlA* **	Paracellular invasion	pldA- 361 pldA-726	AAGAGTGAGGGAAATTCCA GCAAGATGGCAGGATTATCA	385	20	30	95 °C—10 min; 35 cycles: 95 °C—1 min, 45 °C—1 min, 72 °C—2 min; 72 °C—10 min	Zheng et al. [19]
** *cadF* **	Colonization	cadFI-F2B cadFI-R1B	TTGAAGGTAATTTAGATATG CTAATACCTAAAGTTGAAAC	400	20	30	95 °C—10 min; 35 cycles: 95 °C—1 min, 45 °C—1 min, 72 °C—2 min; 72 °C—10 min	Zheng et al. [19]
** *ciaB* **	Intracellular invasion	ciaBI-652 ciaB-1159	TGCGAGATTTTTCGAGAATG TGCCCGCCTTAGAACTTACA	527	20	30	95 °C—10 min; 35 cycles: 95 °C—1 min, 45 °C—1 min, 72 °C—2 min; 72 °C—10 min	Zheng et al. [19]
** *cdtABC* **	Multiplex PCR: Cytotoxin	cdtA-F cdtA-R cdtB-F cdtB-R cdtC-F cdtC-R	CTATTACTCCTATTACCCCACC AATTTGAACCGCTGTATTGCTC AGGAACTTTACCAAGAACAGCC GGTGGAGTATAGGTTTGTTGTC ACTCCTACTGGAGATTTGAAAG CACAGCTGAAGTTGTTGTTGGC	420 531 339	80	20	94 °C—5 min; 30 cycles: 94 °C—1 min, 57 °C—1 min, 72 °C—1 min; 72 °C—5 min	Martinez et al. [20]
** *dnaJ* **	Thermotolerance	dnaJ F dnaJ R	AAGGCTTTGGCTCATC CTTTTTGTTCATCGTT	720	20	20	95 °C—2 min; 30 cycles: 94 °C—1 min, 46 °C—1 min, 72 °C—1 min; 72 °C—5 min	Datta et al. [21]
** *sodB* **	Oxidative stress	sodB F sodB R	ATGATACCAATGCTTTTGGTGATTT TAATACGACTCACTATAGGGCATTTGCATA AAAGCTAACTGATCC	638	20	20	95 °C—2 min; 30 cycles: 94 °C—1 min, 46 °C—1 min, 72 °C—1 min; 72 °C—5 min	Biswas et al. [22]
** *luxS* **	Quorum-sensing	luxS-1 luxS-2	AGGCAAAGCTCCTGGTAAGGCCAA GGATCCGTATAGGTAAGTTCATTTT TGCTCC	1080	50	10	94 °C—3 min; 30 cycles: 94 °C—30 s, 57 °C—1 min; 72 °C—1 min; 72 °C—10 min	Elvers, Park [23]
**HLWL85** **1290**	RAPD-PCR: genetic similarity	HLWL85 1290	ACGTATCTGC GTGGATGCGA	--	10	30	92 °C—2 min; 35 cycles: 92 °C—15 s, 36 °C —1 min; 72 °C—1 min; 1 final cycle at 72 °C—5 min.	Akopyanz et al. [24]
** *ciaB* **	RT-PCR: Invasion		ATATTTGCTAGCAGCGAAGAG GATGTCCCACTTGTAAAGGTG	157	200	4	94 °C—3 min; 45 cycles: 94 °C—15 s, 51 °C—20 s, 72 °C—20 s; 72 °C—3 min	Li et al. [15]
** *dnaJ* **	RT-PCR: Thermotolerance		AGTGTCGAGCTTAATATCCC GGCGATGATCTTAACATACA	117	200	4	94 °C—3 min; 45 cycles: 94 °C—15 s, 51 °C—20 s, 72 °C—20 s; 72 °C—3 min	Li et al. [15]
** *p19* **	RT-PCR: Iron uptake under stress		GATGATGGTCCTCACTATGG CATTTTGGCGTGCCTGTGTA	206	200	4	94 °C—3 min; 45 cycles: 94 °C—15 s, 51 °C—20 s, 72 °C—20 s; 72 °C—3 min	Birk et al. [14]
** *sodB* **	RT-PCR: Oxidative Stress		TATCAAAACTTCAAATGGGG TTTTCTAAAGATCCAAATTCT	170	200	4	94 °C—3 min; 45 cycles: 94 °C—15 s, 51 °C—20 s, 72 °C—20 s; 72 °C—3 min	Birk et al. [14]

**Table 3 ijerph-19-06087-t003:** Prevalence of *Campylobacter* spp. in different meat matrices and forms of commercialization in Brazil in the period from 2014 to 2016.

MATRIX	N	Forms of Commercialization (N)	*Campylobacter* spp. *n*/N (%)	*C. coli* *n*/N (%)	*C. jejuni* *n*/N (%)	TOTAL *n*/N (%)
**Chicken carcasses**	246	Frozen (166) Chilled (80)	8/166 (4.82) 2/80 (2.50) ^a^	9/166 (5.42) 2/80 (2.50) ^a^	19/166 (11.45) 6/80 (7.50) ^b^	36/166 (21.69) 10/80 (12.50) ^I^
**Chicken liver**	138	Frozen (114) Chilled (24)	1/114 (0.87) 1/24 (4.16) ^a^	1/114 (0.87) 1/24 (4.16) ^a^	1/114 (0.87) --- ^a^	3/114 (2.63) 2/24 (8.33) ^II^
**Bovine liver**	246	Frozen (114) Chilled (132)	0 7/132 (5.30) ^a^	0 2/132 (1.51) ^a^	0 2/132 (1.51) ^a^	--- 11/132 (8.33) ^II^
**Minced meat**	138	Chilled (138)	5/138 (3.62) ^a^	0 ^a^	0 ^a^	5/138 (3.62) ^II^
**Pork shank**	138	Chilled (138)	6/138 (4.35) ^a^	5/138 (3.62) ^a^	3/138 (2.17) ^a^	14/138 (10.14) ^II^
**TOTAL**	906	Chicken carcasses (246) Other meat matrix (660)	10/246 (4.07) ^a^ 20/660 (3.03) ^a^	11/246 (4.47) ^a^ 9/660 (25.71) ^ab^	25/246 (10.16) ^b^ 6/35 (17.14) ^b^	46/246 (18.69) ^I^ 35/660 (5.30) ^II^
		Frozen (394) Chilled (512)	9/394 ^a^ 21/512 ^a^	10/394 ^a^ 10/512 ^a^	20/394 ^a^ 11/512 ^a^	39/394 ^I^ 42/512 ^I^
		Total (906)	30/906 (3.31) ^a^	20/906 ^a^	31/906 ^a^	81/906

N = total number of samples; *n* (%) = number of positive samples and percentage. Different numbers in the same column ^I,II^ and letters in the same line ^a,b^ indicate a statistical difference in each variable analyzed (*p* < 0.05, Fisher’s exact test).

**Table 4 ijerph-19-06087-t004:** Antimicrobial resistance in *Campylobacter* strains isolated in the study.

Antimicrobial	Source N = 81	*C. jejuni* n^1^ = 25 n^2^ = 6	*C. coli* n^1^ = 11 n^2^ = 9	*Campylobacter* spp. n^1^ = 10 n^2^ = 20	Total N^1^ = 46 N^2^ = 35
	Chicken carcasses ^1^	14/25 (56%)	8/11 (72.7%)	6/10 (60%)	28/46 (60.9%) ^a^
CIP	Other meat matrices ^2^	2/6 (33.3%)	3/9 (33.3%)	8/20 (40%)	13/35 (37.1%) ^b^
	Total	16/31 (51.6%)	11/20 (55%)	14/30 (46.7%)	41/81 (50.61%)
	Chicken carcasses ^1^	11/25 (44%)	5/11 (45.5%)	4/10 (40%)	25/46 (54.3%)
AMC	Other meat matrices ^2^	2/6 (33.3%)	4/9 (44.4%)	9/20 (45%)	15/35 (42.9%)
	Total	13/31 (41.9%)	9/20 (45%)	13/30 (43.3%)	40/81 (49.4%)
	Chicken carcasses ^1^	11/25 (44%)	2/11 (18.2%)	4/10 (40%)	17/46 (37%)
GEN	Other meat matrices ^2^	2/6 (33.3%)	2/9 (22.2%)	5/20 (25%)	9/35 (25.7%)
	Total	13/31 (41.9%)	4/20 (20%)	9/30 (30%)	26/81 (32.09%)
	Chicken carcasses ^1^	11/25 (44%)	6/11 (54.5%)	2/10 (20%)	19/46 (41.3%)
ERY	Other meat matrices ^2^	1/6 (16,7%)	2/9 (22.2%)	6/20 (30%)	9/35 (25.7%)
	Total	12/31 (38.7%)	8/20 (40%)	8/30 (26.7%)	28/81 (34.6%)
	Chicken carcasses ^1^	11/25 (44%)	2/11 (18.2%)	8/10 (80%)	21/46 (45,7%)
TET	Other meat matrices^2^	4/6 (66.7%)	5/9 (55.5%)	12/20 (60%)	21/35 (60%)
	Total	15/31 (48.4%)	7/20 (35%) ^I^	20/30 (66.7%) ^II^	42/81 (51.9%)
	Chicken carcasses ^1^	6/25 (24%)	4/11 (36.4%)	3/10 (30%)	13/46 (28.3%)
AZM	Other meat matrices ^2^	1/6 (16.7%)	2/9 (22.2%)	6/20 (30%)	9/35 (25.7%)
	Total	7/31 (22.6%)	6/20 (30%)	9/30 (30%)	22/81 (27.2%)

N: total number of strains isolated, N^1^: total number of isolates from chicken carcasses, N^2^: total number of isolates from other meat matrices, n^1^: number of isolates from chicken carcasses of each species, n^2^: number of isolates from other matrices of each species. CIP: Ciprofloxacin, AMC: Amoxillin and clavulanate, GEN: gentamicin, ERY: Erythromycin, TET: Tetracycline, AZM: Azithromycin. *p* < 0.05 in the column ^a,b^ and in the line ^I,II^, Fisher’s exact test.

**Table 5 ijerph-19-06087-t005:** Multidrug resistance profiles of *Campylobacter* isolated from different meat matrices.

Profiles	Number of Profiles	Caracter	Origin	*Campylobacter* spp.	*C. jejuni*	*C. coli*	Total ^1^	Total ^2^ (%)	
P1	1	Susceptible	Carcasses	1	1		2	2 (2.5) ^A^	
			Other meat matrix	-	-		-		
P2 a P7	6	Monoresistance	Carcasses	-	4	4	8	16 (19.8) ^B^	
			Other meat matrix	4	-	4	8		
P8 a P13	6	Co-Resistance	Carcasses	5	8	2	15 ^I^	19 (23.5) ^B^	
			Other meat matrix	2	-	2	4 ^II^		
P14 a P24	11	MDR (3C)	Carcasses	2	11	2	15	22 (27.2)	44 (54.3) ^C^
			Other meat matrix	5	1	1	7		
P25 a P34	10	MDR (4C and 5C)	Carcasses	2	2	3	7	18 (22.2)/	
			Other meat matrix	6	4	1	11		
P35	1	MDR (all)	Carcasses	-	-	-	0 ^I^	4 (4.9)	
			Other meat matrix	3	-	1	4 ^II^		
Total	35	-	Total	30	31	20	81		

P: profile, MDR: multidrug resistance, 3/4/5C: number of antimicrobial classes included in the profiles, Total ^1^: broken down by sample type, Total ^2^: total number of isolates in the respective profile group, ^I,II^ *p* < 0.05 when comparing each profile group between matrix types, ^A,B,C^ *p* < 0.05 when comparing profile groups, Fisher’s exact test.

**Table 6 ijerph-19-06087-t006:** Number and percentage of virulence genes in the 81 strains of *Campylobacter* spp. isolated from meat products.

GENE	Chicken Carcasses-*n*(%)			Total ^1^ N(%)	Other Meat Matrix-*n*(%)			Total ^2^ N(%)	TOTAL
	*C. jejuni*	*C. coli*	*C.* spp.		*C. jejuni*	*C. coli*	*C.* spp.		N(%)
*ciaB*	23 (92.0)	7 (63.6)	7 (70.0)	37 (80.4) ^a^	4 (66.6)	5 (55.5)	4 (20.0)	13 (37.1) ^b^	50 (61.7)
*pldA*	21 (84.0)	8 (72.7)	9 (90.0)	38 (82.6) ^a^	3 (50.0)	2 (22.2)	6 (30.0)	11 (31.4) ^b^	49 (60.5)
*flaA*	19 (76.0)	11 (100.0)	6 (60.0)	36 (78.3) ^a^	2 (33.3)	4 (44.4)	2 (10.0)	8 (22.8) ^b^	44 (54.3)
*cadF*	24 (96.0)	11 (100.0)	9 (90.0)	44 (95.6) ^a^	3 (50.0)	6 (50.0)	9 (45.0)	18 (51.4) ^b^	62 (76.5)
*cdtA*	14 (56.0)	3 (27.3)	4 (40.0)	21 (45.6) ^a^	2 (33.3)	0	1 (5.0)	3 (8.5) ^b^	24 (29.6)
*cdtB*	15 (60.0)	3 (27.3)	4 (40.0)	22 (47.8) ^a^	2 (33.3)	0	1 (5.0)	3 (8.5) ^b^	25 (30.9)
*cdtC*	14 (56.0)	3 (27.3)	4 (40.0)	21 (45.6) ^a^	1 (16.7)	0	1 (5.0)	2 (5.7) ^b^	23 (28.4)
*luxS*	14 (56.0)	0	1 (10.0)	15 (32.6) ^a^	6 (100.0)	5 (55.5)	14 (70.0)	25 (71.4) ^b^	40 (49.4)
*dnaJ*	18 (72.0)	7 (63.6)	5 (50.0)	30 (65.3) ^a^	1 (16.7)	5 (55.5)	7 (35.0)	13 (37.1) ^b^	43 (53.0)
*sodB*	2 (8.0)	0	0	2 (4.4) ^a^	0	0	0	0 ^a^	2 (2.4)
**TOTAL**	**25 (54.3)**	**11 (24.0)**	**10 (21.7)**	**46 (100)**	**6 (17.1)**	**9 (25.7)**	**20 (57.2)**	**35 (100)**	**81 (100)**

*C.* spp.: *Campylobacter* spp.; Total ^1^ refers to isolates from chicken carcasses. Total ^2^ refers to isolates from other breeds. Different letters in ^(a,b)^ the same row indicate *p* < 0.05, Fisher’s exact test.

## Data Availability

Not applicable.

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
