# Peer review of "Agents of Campylobacteriosis in Different Meat Matrices in Brazil"

_ijerph, 2022, doi:10.3390/ijerph19106087_

Round 1
Reviewer 1 Report
The manuscript describe detail investigation of Campylobacter found in food items in Brazil using various techniques. These new information can be used for further risk assessment and should be accepted for publication.
The manuscript can be improved as follow;
- line 74-78. Add details of the sampling design including the sampling frame (How much meat was produced during the sampling period), sample selection method (how was each meat sample selected from all the meat produced) and sample collection (how did you get 150g or 10 g from the whole carcass/organ) for each meat matrix. If sampling was done according to an established surveillance programme, please provide reference.
- line 153 - 165. Add reference for disk diffusion susceptibility testing method (ISO20776-1 is a standard for broth micro dilution test). Add citation for interpretation criteria with reference for classification of bacteria into sensitive and resistance (EUCAST, 2018 is not listed in the references section)
- line 185-186. "Quantitative method" is not described. Please add details in method section
- Table 5. Add column "number of profiles" with number of profile included in each row (Total 35)
- Line 439. "CIP" not "IPC"
- Line 578-580. Add details of the strains not analyzed by RAPD (were they random or specific species from specific matrix only)
- Use 3 decimals significant level throughout the manuscript.
Author Response
- Reviewer #1:
Limitations:
Line 74-78. Add details of the sampling design including the sampling frame (How much meat was produced during the sampling period), sample selection method (how was each meat sample selected from all the meat produced) and sample collection (how did you get 150g or 10 g from the whole carcass/organ) for each meat matrix. If sampling was done according to an established surveillance programme, please provide reference.
Author's response: The sampling information is available in the text on lines 71 to 80 and was determined according to the formula also added in the paper. Since the project was in collaboration with ANVISA, the method for selecting and obtaining samples was determined by ANVISA and in accordance with the current proposal approved by the funding agency (CNPq). We have included in the text that the samples were collected directly from the commercializing markets of the products (Line 73) and therefore we have not included the information about 'How much meat was produced during the sampling period'.
Line 153 - 165. Add reference for disk diffusion susceptibility testing method (ISO20776-1 is a standard for broth micro dilution test). Add citation for interpretation criteria with reference for classification of bacteria into sensitive and resistance (EUCAST, 2018 is not listed in the references section)
Author's response: We thank you for the caveats. The references have been added in Line 172.
Line 185-186. "Quantitative method" is not described. Please add details in method section
Author's response: We apologize for our misunderstanding. The methodology is described in lines 124 to 130.
Table 5. Add column "number of profiles" with number of profile included in each row (Total 35)
Author's response: We add the column with the number of profiles.
Line 439. "CIP" not "IPC"
Author's response: We apologize for the error. The correction has been made (Line 418).
Line 578-580. Add details of the strains not analyzed by RAPD (were they random or specific species from specific matrix only)
Author's response: Information from strains not analyzed by RAPD showed a random pattern and was included in the results, in lines 296-298 references to figure 2 and 326-328 references to figure 3.
Use 3 decimals significant level throughout the manuscript.
Author's response: We standardize all p-values to 3 decimal places. We thank you for the suggestion.
Reviewer 2 Report
- Please include a concluding remark at the end of 'Abstract' section.
- Introduction section of this paper must be improved.
I suggest including the following references either in Introduction or Discussion section:
1) Zhao S, Young SR, Tong E, et al. Antimicrobial resistance of Campylobacter isolates from retail meat in the United States between 2002 and 2007. Appl. Environ. Microbiol. 2010; 76 (24): 7949–56.
2) Kabir SML, Chowdhury N, Asakura M, et al. Comparison of established PCR assays for accurate identification of Campylobacter jejuni and Campylobacter coli. Jpn. J. Infect. Dis. 2019; 72 (2): 81-87. - Campylobacter/C. jejuni/C. coli will be italic throughout the text.
- Resolution of Figure 2 and Figure 3 must be improved.
- Discussion is too long. Please delete unnecessary parts and try to improve this section by mentioning the latest references.
- I suggest to use 60-70 references for this paper.
Author Response
#Reviewer 2:
Please include a concluding remark at the end of 'Abstract' section.
Author's response: We have rewritten the conclusion of the abstract. Sorry for the misunderstandings.
Introduction section of this paper must be improved. I suggest including the following references either in Introduction or Discussion section: 1) Zhao S, Young SR, Tong E, et al. Antimicrobial resistance of Campylobacter isolates from retail meat in the United States between 2002 and 2007. Appl. Environ. Microbiol. 2010; 76 (24): 7949–56.
2) Kabir SML, Chowdhury N, Asakura M, et al. Comparison of established PCR assays for accurate identification of Campylobacter jejuni and Campylobacter coli. Jpn. J. Infect. Dis. 2019; 72 (2): 81-87.
Author's response: The references were inserted in the introduction on lines 43 and 52. Thank you for the suggested improvement
Campylobacter/C. jejuni/C. coli will be italic throughout the text.
Author's response: Thank you for reporting. Adjustments were made throughout the text (marked in red).
Resolution of Figure 2 and Figure 3 must be improved.
Author's response: The images were generated by the GelCompar II program itself. We used an online program to try to improve the quality and we have the image in jpeg that can be inserted with more quality by editors. In annex we insert the images in jpeg.
Discussion is too long. Please delete unnecessary parts and try to improve this section by mentioning the latest references.
Author's response: The discussion was reduced and the amount of references was adjusted, as suggested.
I suggest to use 60-70 references for this paper.
Author's response: The references were adjusted to 78. We were not able to further reduce the number of references due to the number of tests that were done in this study that required the citation of authors that we could not exclude. We reduced the number of references from 133 to 78.
Reviewer 3 Report
The manuscript titled “Agents of campylobacteriosis in different meat matrices in Brazil” concerns the identification of the prevalence of thermophilic species of Campylobacter in meats of different species available on the Brazilian commercial market and to determine the genetic diversity, antimicrobial resistance and virulence potential of the isolates.
General comments
The authors did a lot of tests and researched many factors. However, the present description of the study needs some improvements.
The material and methods should be revised and supplied with additional information.
Regarding discussion, I suggest shortening the section. The present text is difficult to read i.a. because of frequent repetition of the results (like “The highest prevalences of Campylobacter were found in chicken carcasses and pork 334 shank (36.0% and 10.1%, respectively)”) as well as extensive comparison with other results from the literature. It would be more valuable to have an extended discussion of the relevance of the results. In conclusion: shorten the simple comparison (i.e. your own results versus other results) and extend the substantive discussion.
The conclusion needs linguistic reformulation and more reference for prevalence in Brazil and answers whether the aim was realised:
• panoramic analysis of the occurrence of Campylobacter in meats marketed in Brazil,
• the prevalence of the genus,
• main species of public health importance,
• antimicrobial resistance,
• virulence potential and genetic diversity.
Line 74-76 Please, add more details regarding “sanitary inspection, qualified for domestic trade and/or export”
Line 79: Please explain what is "Pesp"
Line 80: The cited article are not placed on the reference list.
Line 131: References mentioned in the table should be listed on the reference list
Line 187 I suggest not writing 123 ± 190 CFU/g, only average and min and max
Line 196 What kind of "processing" do the authors mean?
Line 197 What do the authors mean by "marketed frozen"? In section 2.1 there is no information about buying the frozen product...
Line 202 This information should be in section 2.1
Line 334 Should be Campylobacter spp. and in other lines when the information refers to Campylobacter spp.
Author Response
#Reviewer 3:
The material and methods should be revised and supplied with additional information.
Author's response: The methodology were adjusted to improve the explanation about sampling and quantitative analysis, in lines 71 to 80 and 124 to 130. If the reviewer has something specific, we can change the text further.
Discussion - I suggest shortening the section. The present text is difficult to read i.a. because of frequent repetition of the results (like “The highest prevalences of Campylobacter were found in chicken carcasses and pork 334 shank (36.0% and 10.1%, respectively)”) as well as extensive comparison with other results from the literature. It would be more valuable to have an extended discussion of the relevance of the results. In conclusion: shorten the simple comparison (i.e. your own results versus other results) and extend the substantive discussion.
Author's response: The discussion was revised and expanded with a focus on substantive discussion, as suggested. We have reduced a total of 78 lines of discussion. We thank you for your considerations.
The conclusion needs linguistic reformulation and more reference for prevalence in Brazil and answers whether the aim was realised:
• panoramic analysis of the occurrence of Campylobacter in meats marketed in Brazil,
• the prevalence of the genus,
• main species of public health importance,
• antimicrobial resistance,
• virulence potential and genetic diversity.
Author's response: The conclusion has been reworded accordingly.
Line 74-76 Please, add more details regarding “sanitary inspection, qualified for domestic trade and/or export
Author's response: We have included information regarding the three types of inspection systems present in the country (Municipal and State - which serve the domestic market - and Federal - which also serves the foreign market), in lines 69 to 71.
Line 79: Please explain what is "Pesp"
Author's response: We included it in the text on line 79.
Line 80: The cited article are not placed on the reference list.
Author's response: References from Table 1 were inserted and marked in red.
Line 131: References mentioned in the table should be listed on the reference list
Author's response: References from Table 2 were inserted and marked in red.
Line 187 I suggest not writing 123 ± 190 CFU/g, only average and min and max
Author's response: The suggestion was accepted and the text was corrected, the deviation was removed (Line 199).
Line 196 What kind of "processing" do the authors mean?
Author's response: The "processing" is described in section 2. Materials and methods. And it corresponds to step/stage 2. We have changed the text in line 208 to make it clearer.
Line 197 What do the authors mean by "marketed frozen"? In section 2.1 there is no information about buying the frozen product.
Author's response: The sentence has been rewritten.
However, slow subsequent freezing (in-home) of the positive samples did not maintain Campylobacter viability, since in the second stage (samples kept frozen for 30 days after processing), we reisolated the bacteria only in 2/46 chicken carcass samples originally sold frozen (one C. jejuni and one Campylobacter spp.).
Line 202 This information should be in section 2.1
Author's response: The excerpt "The samples were obtained from a total of 31 producers, of which 14 were representative brands from the country's major chicken carcass markets..." has been reorganized at 71-73 (section 2.1). We thank you for your verification.
Line 334 Should be Campylobacter spp. and in other lines when the information refers to Campylobacter spp.
Author's response: This sentence refers to the general that was identified, both of C. jejuni, C. coli and Campylobacter spp. We changed the text to Campylobacter genus on line 351.
Round 2
Reviewer 1 Report
The manuscript has improved significantly. I would like to suggest the following edits:
- Line 76-80 - remove the formula.
- Line 94-95, Table3 legend - Fisher's exact test (not Fischer test)
- Table 3 - using letters to indicate significant different between both rows and column can be confusing. Using letter for between row comparison only and use number for between column comparison may be easier to understand.
- Figure 1 - Please add description of "internal" and "external" consumer market (indoor/outdoor?) in the method section.
- Table 4, 5 - Fisher's exact test. Again using * for different comparison can be confusing. Please use different letters or numbers.
Author Response
Response Letter – Manuscript ID: IJERPH – 1670566
Title: “Agents of campylobacteriosis in different meat matrices in Brazil”
Comments from the reviewer:
- Reviewer #1:
The manuscript has improved significantly. I would like to suggest the following edits:
Author's response: We thank you for your collaboration.
Line 76-80 - remove the formula.
Author's response: We removed.
Line 94-95, Table3 legend - Fisher's exact test (not Fischer test)
Author's response: We modified.
Table 3 - using letters to indicate significant different between both rows and column can be confusing. Using letter for between row comparison only and use number for between column comparison may be easier to understand.
Author's response: We modified.
Figure 1 - Please add description of "internal" and "external" consumer market (indoor/outdoor?) in the method section.
Author's response: We made the inclusion in lines 71 and 72.
Table 4, 5 - Fisher's exact test. Again using * for different comparison can be confusing. Please use different letters or numbers.
Author's response: We modified.